# *Mycobacterium tuberculosis* *whiB3* and Lipid Metabolism Genes Are Regulated by Host Induced Oxidative Stress

**DOI:** 10.3390/microorganisms10091821

**Published:** 2022-09-11

**Authors:** Omar M. Barrientos, Elizabeth Langley, Yolanda González, Carlos Cabello, Martha Torres, Silvia Guzmán-Beltrán

**Affiliations:** 1Departamento de Investigación en Microbiología, Instituto Nacional de Enfermedades Respiratorias Ismael Cosío Villegas, Calzada de Tlalpan 4502, Sección XVI, Alcaldía de Tlalpan, Mexico City 14080, Mexico; 2Departamento de Investigación Básica, Instituto Nacional de Cancerología, Mexico City 14080, Mexico; 3Departamento de Virologia y Micologia, Instituto Nacional de Enfermedades Respiratorias Ismael Cosío Villegas, Mexico City 14080, Mexico; 4Laboratorio de Inmunobiología de la Tuberculosis, Instituto Nacional de Enfermedades Respiratorias Ismael Cosío Villegas, Mexico City 14080, Mexico

**Keywords:** *Mycobacterium tuberculosis*, oxidative stress, WhiB3, lipid metabolism

## Abstract

The physiological state of the human macrophage may impact the metabolism and the persistence of *Mycobacterium tuberculosis*. This pathogen senses and counters the levels of O_2_, CO, reactive oxygen species (ROS), and pH in macrophages. *M. tuberculosis* responds to oxidative stress through WhiB3. The goal was to determine the effect of NADPH oxidase (NOX) modulation and oxidative agents on the expression of *whiB3* and genes involved in lipid metabolism (*lip-Y*, *Icl-1*, and *tgs-1*) in intracellular mycobacteria. Human macrophages were first treated with NOX modulators such as DPI (ROS inhibitor) and PMA (ROS activator), or with oxidative agents (H_2_O_2_ and generator system O_2_^•−^), and then infected with mycobacteria. We determined ROS production, cell viability, and expression of *whiB3*, as well as genes involved in lipid metabolism. PMA, H_2_O_2_, and O_2_^•−^ increased ROS production in human macrophages, generating oxidative stress in bacteria and augmented the gene expression of *whiB3*, *lip-Y*, *Icl-1*, and *tgs-1*. Our results suggest that ROS production in macrophages induces oxidative stress in intracellular bacteria inducing *whiB3* expression. This factor may activate the synthesis of reserve lipids produced to survive in the latency state, which allows its persistence for long periods within the host.

## 1. Introduction

Tuberculosis remains a global health problem. Close to one-third of the world’s population is infected, but only 5–10% develop active disease [1]. The causative agent, *Mycobacterium tuberculosis*, is capable of growing actively or remaining dormant for years within its host [2]. Efforts to contain tuberculosis have been restricted by the lack of an effective vaccine, as well as the high incidence of multi-drug resistant strains. This makes it necessary to search for new drug targets leading to better treatment options. In this respect, a clearer understanding of the mechanisms involved in stress resistance in the host environment would supply important leads for new therapies.

The success of mycobacteria rests on their ability to sense extreme conditions within their host, such as nutrient limitation, acid pH, hypoxia, and redox stress. Several studies have revealed that *M. tuberculosis* possesses sophisticated mechanisms to continuously monitor and mount proper responses against host-generated stresses. Mycobacteria activate several transcriptional regulators in response to adverse conditions. For example, it controls various regulators involved in counteracting oxidative stress, such as sigma factors (-H and -E), the two-component systems (DOSR/S/T and SenX-RegX), and transcriptional regulators such as MosR, HpoR, among others [3,4,5,6,7]. Additionally, *M. tuberculosis* encodes seven WhiB (WhiB1-WhiB7) redox-sensing transcription factors, which all contain a characteristic Fe–S cluster [8]. The WhiB proteins in mycobacteria have diverse functions which include maintaining redox homeostasis, regulating secretion systems, virulence, antibiotic resistance, and reactivation from dormancy. WhiB3 is the most studied protein in this family. This factor influences mycobacterial pathogenesis by avoiding phagosomal maturation and modulating the cell cycle [9,10,11,12]. Furthermore, WhiB3 protects from acidic pH encountered inside cells by sustaining the mycothiol redox system [13].

The WhiB3 iron-sulfur cluster has been shown to be sensitive to reactive oxygen and nitrogen species [14]. Its different oxidation status permits sensing and maintaining redox homeostasis. WhiB3 also induces early granuloma development in an in vitro model of human peripheral blood mononuclear cells [12]. Furthermore, WhiB3 has been associated with lipid metabolism during dormancy, controlling lipid degradation and the synthesis of reserve lipids, such as triacylglycerols (TAG), to form inclusion bodies. Additionally, WhiB3 regulates the production of lipids that cause inflammation, such as PAT, DAT, SL-1, and PDIM in virulent strains [10].

Mycobacteria can survive in hostile microenvironments within macrophages, inducing antioxidant systems such as alkylhydroperoxide reductase (AhpC), catalase/peroxidase (KatG), peroxiredoxin (AhpE), thioredoxin reductase (Tpx), and superoxide dismutase (SodA) [15]. SodA is a metalloprotein that catalyzes the dismutation reaction of O_2_^•−^ into H_2_O_2_ and molecular O_2_ [16].

During infection, *Mycobacterium* uses its own lipases to hydrolyze host lipids. Lipases hydrolyze ester bonds in long-chain acylglycerols to liberate fatty acids and glycerol [17]. This enzyme is localized on the cell surface and its overexpression reduces TAG reserves [18].

TAG accumulation is an essential event in dormancy. TAG synthesis in *M. tuberculosis* is regulated by triacylglycerol synthase 1 (Tgs1) [19]. Tgs-1 is the final enzyme involved in the synthesis of mycobacterial endogenous TAGs. It catalyzes the incorporation of fatty acid into a diacylglycerol molecule. Deletion of its gene causes a decrease in the amount of TAG stored during hypoxia-induced dormancy [19].

The *icl-1* gene encodes isocitrate lyase (Icl), which is the principal enzyme in the glyoxylate cycle [20]. Icl is necessary for the use of fatty acids and is important for latency in tuberculosis [21]. *Icl-1* expression increases significantly under hypoxia and can use fatty acids as sole carbon and energy sources, through the glyoxylate cycle, when the carbon source is reduced [22].

Infected human macrophages induce defense mechanisms such as oxidative stress to eliminate the pathogen. This condition causes the mycobacteria to change their metabolism to survive and persist in the host. WhiB3 has been associated with increased metabolism of reserve lipids to survive dormancy in the host under adverse conditions, such as oxidative stress. The main goal of this project was to determine the effect of NADPH oxidase modulation and oxidative agents on the expression of whiB3 and genes involved in lipid metabolism (*lip-Y*, *Icl-1*, and *tgs-1*) of *M. tuberculosis.*

## 2. Materials and Methods

### 2.1. Strains and Culture Conditions

*M. tuberculosis* H37 Ra was purchased from the American Type Culture Collection (ATCC 25177, Rockville, MD, USA). Mycobacteria were cultured at 37 °C in 7H9 broth (Difco, Detroit, MI, USA) supplemented with 0.2% glycerol and 0.05% tween 80 (Sigma-Aldrich, St. Louis, MO, USA) and ADC (Beckton Dickinson, BD, San Jose, CA, USA) or on solid Middlebrook 7H10 medium (Difco) supplemented with OADC (BD) and 0.5% glycerol.

### 2.2. Human Macrophage Differentiation

Monocyte-derived macrophages (MDMs) were differentiated from monocytes obtained from peripheral blood mononuclear cells from healthy donors at the Instituto Nacional de Enfermedades Respiratorias (INER), under approval from the Institutional Ethical Review Board. Buffy coats were centrifuged, and peripheral blood mononuclear cells were diluted 1:4 with RPMI 1640 (Lonza, Walkersville, MD, USA). Cells were gradient separated by layering onto a lymphocyte separation solution (Lonza) [23]. Then, monocytes (MN) were isolated from peripheral blood mononuclear cells by positive selection using anti-human CD14 coupled to magnetic beads using the MACS Miltenyi purification system (Miltenyi Biotech, Auburn, CA, USA). Cell viability was measured by Trypan blue exclusion and found to be higher than 98.5%. MN concentration was adjusted to 10^6^ cells/mL and the cells were incubated in RPMI 1640 supplemented with 10% heat-inactivated human serum (Valley Biomedicals, VA, USA), and 200 mM L-glutamine (Lonza) on 24-well plates for 7 days for differentiation into macrophages (MDM). The differentiated cells presented <85% of CD206 receptor and 50% of cells showing CD14 expression [24,25]. 

### 2.3. Bronchoalveolar Cell Culture

Bronchoalveolar cells were obtained from lavage remnants of healthy donors. Briefly, remnants were centrifuged, and cells were resuspended in RPMI medium. More than 90% of the bronchoalveolar cells were alveolar macrophages (AM). Cells were maintained in RPMI 1640 supplemented as described above at 37 °C and 5% CO_2_.

### 2.4. Modulation of NOX Activity in Infected Macrophages

Phorbol 12-myristate 13-acetate (PMA, Sigma-Aldrich) activates phosphatidyl inositol 3 kinase promoting the assembly of NOX, and increasing its activity [26]. While Diphenyliodonium chloride (DPI, Sigma-Aldrich) binds to the NOX2 subunit and blocks electron transfer, thereby preventing ROS generation [27]. PMA (750 ng/mL) and DPI (20 µM) were added 1 h before infection and maintained for 2 hours after. We quantified viability and ROS production of the infected macrophages, and gene expression in intracellular bacteria.

### 2.5. Macrophage Infection and Treatment under Oxidative Conditions

Before infection, mycobacteria were disaggregated to achieve a single-cell suspension [28]. Then, 2 × 10^6^ macrophages were infected using human serum-opsonized mycobacteria at a multiplicity of infection (MOI) of 1:15 in supplemented RPMI without antibiotics and incubated for 1 h at 37 °C. The MDMs were then washed with RPMI to remove non-internalized bacteria and fresh medium was added with the different treatments, after which the cells were cultured for 2 more hours.

The macrophages infected with *M. tuberculosis* were treated with 5 mM H_2_O_2_ and O_2_^•−^ generator system using a xanthine-xanthine oxidase (X/XO, Sigma-Aldrich) reaction. Xanthine oxidase catalyzes xanthine to uric acid and O_2_^•−^. Oxidants were added 15 min prior to infection and up to 2 h after. Then, we quantified the viability and presence of ROS, as well as gene expression in intracellular bacteria.

### 2.6. Quantification of Macrophage Viability

0.3 × 10^6^ MN in 300 µL per well, in 48-well plates and incubated for seven days to allow cell differentiation. Cells were then infected and treated under different conditions, as mentioned above. Subsequently, macrophage viability was determined by the release of lactate dehydrogenase, as per the manufacturer’s specifications (CytoTox 96 Non-Radioactive Cytotoxicity Assay, Promega, Madison, WI, USA).

### 2.7. ROS Detection in Infected Macrophages

NBT (Nitro-tetrazolium Blue, Sigma-Aldrich) is a yellow, water-soluble tetrazolium salt. In the presence of ROS, NBT is reduced to formazan, a colorful substance insoluble in water. To measure ROS, 0.3 × 10^6^ cells were seeded per well in a 48-well plate. After infection and the treatments described, NBT was added at 1 mg/mL and incubated for one hour. The medium was discarded, and cells were washed first with 70% methanol and then with 100% methanol. Finally, the crystals were dissolved in 150 µL of 2 M KOH and 175 µL of DMSO, and the solution was read at 590 nm. The results were reported as fold change in ROS production compared to control (cells infected without treatment).

### 2.8. Stress Conditions in Mycobacterial Culture

Mycobacteria were grown in roller bottles in 1000 mL of supplemented Middlebrook 7H9. Cultures were grown for 7 days and then divided into 50-mL aliquots, returned to the incubator at 37 °C for 1 h to equilibrate, and subjected to various stress conditions for 2 h. Oxidant conditions were as follows: 5 mM hydrogen peroxide (H_2_O_2_, Sigma-Aldrich) and superoxide anion (O_2_^•−^) generator system by xanthine-xanthine oxidase (X/XO, Sigma-Aldrich) reaction. Subsequently, we evaluated different parameters in the different conditions and bacterial survival was determined by monitoring bacterial viability quantified as colony-forming units and reported as percent bacterial survival.

### 2.9. Total RNA Extraction

Total RNA was obtained from cell lysates by disruption with tiny glass beads [29]. First, bacteria were lysed with lysozyme (Sigma-Aldrich, 20 mg/mL) and proteinase K (Sigma-Aldrich, 2 mg/mL) solution and incubated for 10 min at 37 °C. Then, 600 µL of RLT buffer (Qiagen, Hilden, Germany) was added to ensure bacterial lysis. The samples were shaken in a Fast Prep Homogenizer (MP Biomedicals, Santa Ana, CA, USA) at a speed of 6.5, 2 cycles of 30 s and were then centrifuged at 8000× *g* for 1 min (Eppendorf, Hamburg, Germany) to eliminate cell debris. Total RNA was obtained with the RNeasy system (Qiagen, Hilden, Germany) according to the manufacturer’s specifications. Total RNA was quantified by spectrophotometry and stored at −70 °C.

### 2.10. Bacterial Gene Expression

A total of 500 ng of total RNA was used for each sample derived from intracellular mycobacteria and 200 ng for the samples that were obtained from pure cultures in a final reaction volume of 20 µL. Briefly, cDNA was synthesized using the iScript cDNA Synthesis kit (Bio-Rad, Hercules, CA, USA). The relative expression values were estimated using ΔΔCT analysis, with the ribosomal 16S value serving as an internal control. According to the in-silico analysis, the sequences of the genes of interest are identical between *M. tuberculosis* H37 Ra and Rv strains. We used oligonucleotide sequences previously reported for the virulent strain. The oligonucleotide sequences used are listed in Table 1.

### 2.11. Statistical Analysis

Statistical analysis and graphics were carried out using GraphPad Prism version 9.4.1 software (San Diego, CA, USA). We performed Wilcoxon’s one-sample test to establish differences from median values = 1 or 100. Significance was established at *p* < 0.05.

## 3. Results

### 3.1. M. tuberculosis whib3 Expression from Pure In Vitro Cultures and after Macrophage Infection

*whiB3* expression in intracellular mycobacteria from MDM and AM increased 200 and 202-fold, respectively, compared to *M. tuberculosis* cultivated in vitro under aerobic conditions during the mid-exponential growth phase (7 days) (Figure 1).

### 3.2. Effect of ROS on whiB3 and sodA Expression in Intracellular Mycobacteria

Superoxide dismutase A (*sodA*) secreted by *M. tuberculosis*, converts ROS generated by host macrophages to H_2_O_2_. In order to associate the expression of *M. tuberculosis-whiB3* with ROS production, we evaluated the effect of ROS induction by PMA or ROS inhibition by DPI on *M. tuberculosis whiB3* and *sodA* expression in the context of infection in MDM (Figure 2A–D). Treatment with PMA (750 ng/mL) and DPI (20 μM), which promote activation and inhibition of NOX, respectively, did not affect the viability of *M. tuberculosis*-infected macrophages (Figure 2A). PMA increases ROS production 2.3 and 2.4-fold (Figure 2C) in uninfected and infected macrophages, respectively, inducing an increase in *whiB3* expression of 2.1 times (Figure 2D) and of *sodA* expression of 1.6 times (Figure 2B) in intracellular bacteria. Conversely, DPI, as expected, decreased ROS levels as well as *sodA* and *whiB3* expression in intracellular *M. tuberculosis* (Figure 2B,D).

### 3.3. Effect of Oxidants on whiB3 and sodA Expression in Intracellular Mycobacteria

The H_2_O_2_ and O_2_^•−^ are released by macrophages as their initial defense system to eliminate intracellular pathogens. We evaluated the direct effect of H_2_O_2_ and the O_2_^•−^ generator system (X-XO) on intracellular mycobacterial *whiB3* expression. First, we determined oxidative conditions that would not affect macrophage viability (Figure 3A) but increased the levels of ROS. H_2_O_2_ and O_2_^•−^ increase ROS production by 4.9 and 4.8-fold, respectively, in uninfected macrophages and 4.4 and 5.6-fold in infected macrophages (Figure 3C). Additionally, we observed an increase in *sodA* and *whiB3* expression of 3.2 and 4.3-fold after treatment with H_2_O_2_ and O_2_^•−^, respectively (Figure 3B,D).

### 3.4. Effect of Oxidants on Genes Involved in Lipid Metabolism in Intracellular Mycobacteria

In order to determine the effects of oxidizing agents on lipid metabolism in *M. tuberculosis*, and therefore their possible effect on inducing dormancy, we evaluated the expression of *tgs-1*, *lip-Y* and *icl-1* genes. Treatment of intracellular *M. tuberculosis* with PMA increased *lip-Y* and *tgs-1* 1.8 and 1.9-fold, respectively, (Figure 4A,C). Treatment with H_2_O_2_ increased them 2.1 and 2.9-fold, while the O_2_^•−^generator system treatment promoted a 2.9 and 2.4-fold increase (Figure 4B,D). Surprisingly, DPI slightly increased *lip-Y* expression by 1.4-fold (Figure 4A). Regarding *icl-1*, PMA induced a 1.5-fold increase, H_2_O_2_, a 1.6-fold increase and O_2_^•−^, a 2.9-fold increase (Figure 4F), whereas DPI caused a decrease in 0.4-fold in *M. tuberculosis* (Figure 4E).

### 3.5. Effect of Oxidizing Conditions on Bacterial Survival and Gene Expression of M. tuberculosis

To assess whether the *whiB3*, *sodA*, *lip-Y*, *tgs-1* and *icl-1* mycobacterial genes were induced by ROS and not by other molecules produced by macrophages, the oxidants were added directly to the pure *M. tuberculosis* cultures. The H_2_O_2_ and O_2_^•−^ generator system did not affect bacterial viability (Figure 5A) and increased expression of *sodA* by 2 and 2.3-fold, respectively (Figure 5B). Furthermore, both conditions significantly increased *whiB3* expression 3.5 and 3.8-fold (Figure 5C), compared to the untreated cultures.

The lipid metabolic genes *lip-Y* and *tgs-1* expression significantly increased in the presence of H_2_O_2_ and O_2_^•−^ generator system by 2.9 and 3.1-fold, respectively (Figure 5D,E). While *icl-1* augmented its expression only in H_2_O_2_ by 3.2-fold (Figure 5F).

### 3.6. The Induction of whiB3 Expression May Be Controlled by Various Transcription Factors

To gain more knowledge about *whiB3* regulation, we used a bioinformatic approach to look at its promoter sequence. We observed that the upstream region of the *whiB3* promoter shows 70% identity with the consensus binding sequence for MosR, as shown in Table 2, (Table 2). MosR is an oxidation-sensing transcriptional regulator. In redox equilibrium, MosR represses transcription. However, under oxidative stress, MosR dissociates from DNA enabling its own transcription and possibly that of other genes [3,31]. This suggests a possible role for MosR in regulating whiB3 expression under our conditions.

## 4. Discussion

Mycobacteria sense the microenvironment to favor an active or latent infection in the host and several regulatory systems are activated during oxidative conditions [12,32,33,34]. However, the mechanism by which *M. tuberculosis* induces dormancy through oxidative stress is not fully understood. Here, we evaluate the expression of *whiB3*, which is sensitive to oxidation by ROS and promotes redox homeostasis. *whiB3* gene expression is increased in infected macrophages and mouse lungs in early-stage infection, suggesting that conditions within macrophage phagosomes are sufficient for high induction [35,36,37,38,39]. We demonstrated that intracellular mycobacteria from infected human MDM and AM express high levels of *whiB3* compared with pure culture in exponential growth. This suggests an essential role for WhiB3 protein during mycobacterial infection. 

We used an infection model where we analyzed the role of increased ROS production on the induction of *whiB3* and lipid catabolic and anabolic gene expression. PMA activates NOX, increasing ROS production in human macrophages [40]. We confirmed that PMA induced ROS production, generating oxidative stress. The *M. tuberculosis*-infected macrophages treated with the O_2_^•−^ system generator or H_2_O_2_ showed a significantly increased *sodA* expression in intracellular bacteria, showing that the higher expression of this enzyme is an indicator of oxidative stress [41]. In contrast, DPI-treated infected cells did not have increased ROS levels nor *sodA* expression in mycobacteria.

Mycobacteria can survive in a hostile microenvironment in macrophages by inducing the antioxidant system [41]. SodA is a constitutive protein in *M. tuberculosis*. Still, its expression can increase under oxidative stress and has been shown to have an approximately 100-fold increase and export 350-fold more enzyme than *M. smegmatis*, a free-living bacterium [16]. The release of SodA during infection could neutralize reactive oxygen molecules before they reach the mycobacterial cell wall. Furthermore, a SodA inhibitor (diethyldithiocarbamate) increases mycobacterial survival in murine splenic macrophages, suggesting that this enzyme probably contributes to the long-term survival of mycobacteria, in vivo [42].

All oxidizing conditions used in this study increased *whiB3* expression, suggesting that ROS controls *whiB3* gene expression. Additionally, Mehta and Singh, 2019, reported that the *whiB3* deleted (Δ*whiB3*) mycobacterial mutant is more acutely sensitive to oxidant and nitrosative agents showing lower survival. Furthermore, WhiB3 modulates the formation of granulomas after the interaction of mycobacteria with human peripheral blood mononuclear cells [12], suggesting an essential role during the early stages of infection. 

Lipid metabolism is essential for survival because *Mycobacterium* needs to degrade host lipids to synthesize its own lipids [43]. *lipY*, *tgs-1*, and *icl-1* gene expression may be crucial in the transition from active growth to a dormancy state generated during infection. It has been shown that hypoxia in intracellular bacteria promotes an increase in expression of *tgs-1* and *icl-1* just before initiating dormancy, while *lipY* increases during the mid-phase of the dormant stage [44]. *Mycobacterium* can adapt quickly under adverse conditions, modifying lipid metabolism [45]. The observed increase in *tsg-1*, *icl-1*, and *lipY* expression under our conditions is supported by studies showing that a Δ*whiB3* mutant strain shows a differential lipid profile compared to the wild-type strain [10]. This mutant presented the absence of lipids associated with pathogenesis, such as DAT, SL-1 and PAT, and the increase in PDIM and TAG, suggesting WhiB3 is involved in modulating lipid anabolism caused by oxidative stress [10].

During infection, *M. tuberculosis* interacts with the host cell and is exposed to adverse situations. An early step for survival is to catabolize host lipids and synthesize TAG to form intrabacterial lipid inclusions [46]. Transcriptional studies showed high expression of catabolic genes such as lipases (lip-F, -H. -N, -X, and -Y) [30], suggesting an essential function in host lipid degradation. They also showed high expression of lipid anabolic genes, such as *icl*, *tgs-1*, and *tgs-2* for TAG synthesis, to then produce lipid inclusions [30]. Furthermore, mycobacteria isolated from the sputum of volunteers with tuberculosis showed intrabacterial lipid inclusions without a replicative state. This condition may intensify tuberculosis transmission from person to person because mycobacteria are more resistant to adverse conditions [47,48]. The bacterial dormant state in a human granuloma model showed intrabacterial lipid inclusions and resistance to rifampicin [49], emphasizing the importance of lipid metabolism in mycobacterial success.

The intrabacterial lipid inclusion, change in the cell wall, increase their cell size and stopping division, are the hallmarks of dormant bacilli [50]. It is likely that WhiB3 has an essential role in activating dormancy under oxidative stress conditions, leading to latent tuberculosis. High levels of lipids within bacteria and in host cells maintain the dormant state in mycobacteria. In contrast, a significant reduction in lipids in both symbionts, generates a change in metabolism and rapid bacterial division, reactivating its growth [50], supporting the essential role of lipids for both latency and reactivation of tuberculosis.

We evaluated the effect of oxidizing agents in mycobacteria outside the context of infection. The treatment with the O_2_^•−^ system generator and H_2_O_2_ did not affect bacterial survival but generated oxidative stress, as witnessed by significantly augmented *sodA* expression. High levels of Whib3 promote the expression of *sodA*. A transcriptomic analysis of *M. tuberculosis H37Rv* shows that WhiB3 modulates the expression of *sodA* and other antioxidant enzymes in mycobacteria, maintaining redox homeostasis [9,10].

Our results show that oxidative conditions induce *whiB3* expression in pure culture and intracellular bacteria, suggesting that WhiB3 is a general response to permit survival under adverse conditions. We hypnotize that oxidative stress may activate a signal through MosR to induce *whiB3* expression; then, the WhiB3 protein senses and responds to oxidative stress. Previous reports showed that the 4Fe-4S cluster of WhiB3 interacts with RNS and ROS [12]. Additionally, the *ΔwhiB3* mutant adversely affects mycobacterial survival under acidic pH [13]. Similarly, previous studies suggest that *Mycobacterium* harbors acidic stress-responsive factors such as PhoP and Rex3 also providing cross-protection against ROS, RNS, hypoxia, and lysosomal hydrolases [12,32,51].

In Figure 6, we propose a possible effect of oxidants on gene expression of free-living *M. tuberculosis* cultivated in 7H9 and of intracellular mycobacteria. First, the superoxide O_2_^●^^−^ and H_2_O_2_ damage the cell wall and membrane, generating lipoperoxidation. O_2_^●^^−^ is converted into H_2_O_2_. This molecule is electrically neutral and crosses membranes, increasing its concentration in the cell and causing an imbalance of the redox state. MosR regulator senses oxidative stress and induces *whiB3* expression. The constant oxidative condition in the microenvironment favors the [4Fe-4S] cluster oxidation of WhiB3 that permits binding to the promoters of its target genes, generating a metabolic change that leads to the state of dormancy.

## 5. Conclusions

WhiB3 is a transcription factor that coordinates survival in response to various adverse conditions such as hypoxia, low pH, ROS, and RNS in vitro and in the context of infection. ROS production in macrophages induces oxidative stress in intracellular bacteria provoking *whiB3* expression. This transcriptional regulator may activate the degradation of host lipids by inducing *lip-Y*, and increasing lipid synthesis inducing *icl-1* and *tgs-1* leading to the accumulation of intrabacterial lipids that allows bacteria to survive in the dormancy state, resisting antibiotic treatment and persisting for long periods in the host.

## Figures and Tables

**Figure 1 microorganisms-10-01821-f001:**
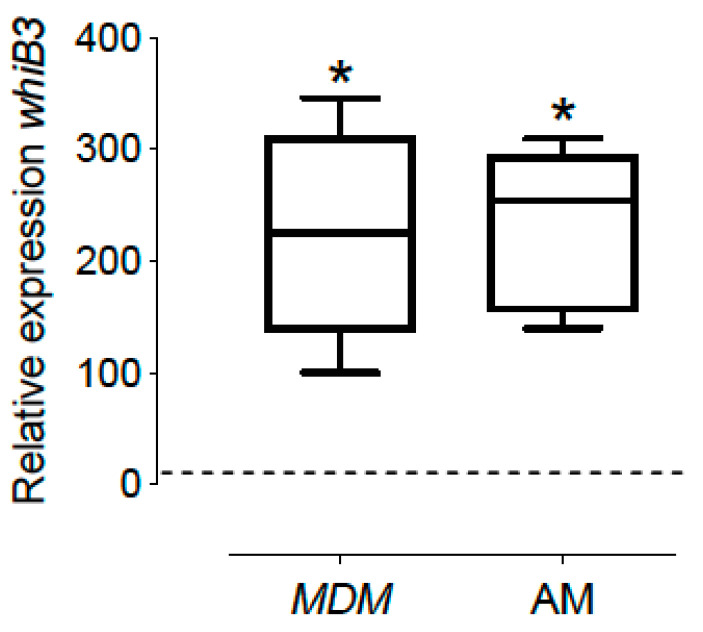
*whib3* expression of *M. tuberculosis* in intracellular bacteria in MDM and AM compared to aerobic in vitro 7H9 growth during mid-exponential growth phase (control, dashed line). Total RNA was isolated and real-time qRT-PCR was performed. Relative expression levels of whiB3 were estimated using the RNA 16S transcript as internal control for normalization of RNA amounts; expression levels from an in vitro culture were used as control. MDM: Monocyte-derived macrophages and AM: alveolar macrophages. The data show the maximums and minimums by quartiles. * *p* < 0.05 significant difference compared to the expression of control (bacteria cultivated in vitro), *n* = 6. The dashed horizontal line represents the median, and the Wilcoxon test was used to compare against the control.

**Figure 2 microorganisms-10-01821-f002:**
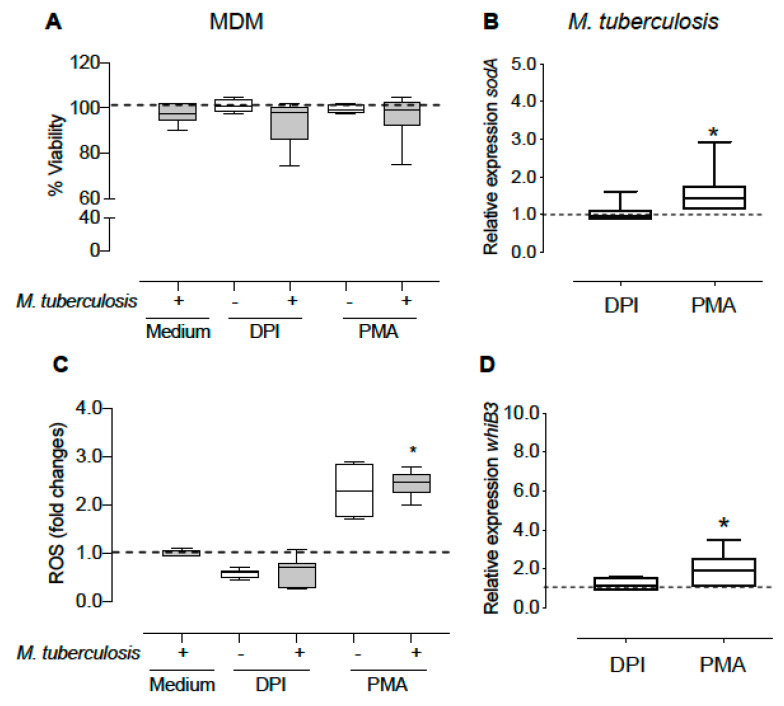
Effect of NOX activity modulation on gene expression of intracellular mycobacteria. The macrophages were treated with PMA (750 ng/mL) or DPI (20 µM) for 1 h. Subsequently, they were infected with *M. tuberculosis* (MOI 1:15) in the presence of PMA or DPI and incubated for another hour. The non-phagocytized bacteria were eliminated and fresh medium with PMA or DPI was added for 2 more hours. We detected cell viability (**A**) and ROS levels (**C**) in MDM. Furthermore, expression levels of *whiB3* and *sodA* (**B**,**D**) were determined in intracellular bacteria. The data show the maximums and minimums by quartiles. * *p* < 0.05 significant difference compared to the expression of uninfected macrophages (**A**,**C**) or intracellular bacteria without treatment (**B**,**D**), *n* = 4–7. The Wilcoxon test was used to compare to the control. In panels (**A**,**C**), the dashed horizontal line represents the median of the uninfected macrophages without treatment, and panels (**B**,**D**), represents the median of the intracellular bacteria without treatment.

**Figure 3 microorganisms-10-01821-f003:**
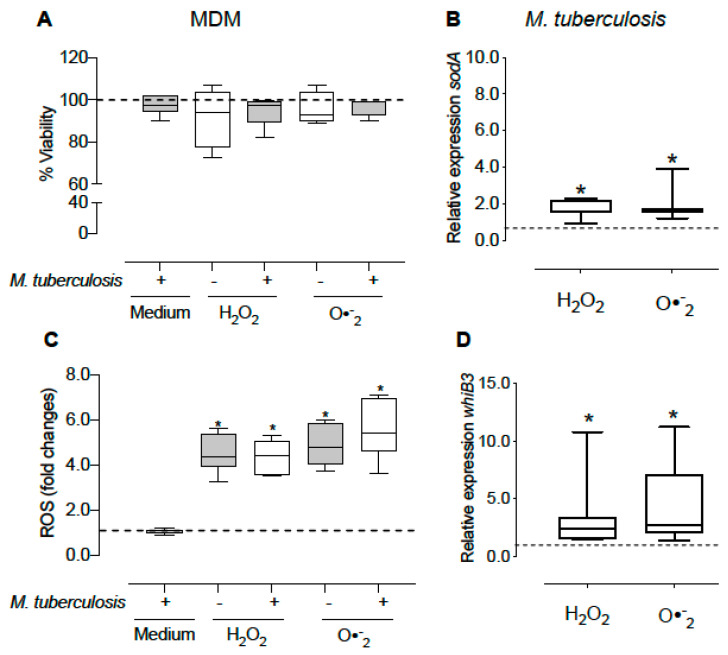
Effect of oxidizing agents on *whiB3* and *sodA* gene expression of intracellular mycobacteria. The MDM were treated with 5 mM H_2_O_2_ and O_2_^•−^ generator system for 15 min, and then infected with *M. tuberculosis* H37Ra (MOI 1:15) in the presence of these oxidants for 1 h. Non-phagocytized bacteria were removed, and fresh medium plus oxidants was added for 2 h. We detected cell viability (**A**) and ROS levels (**C**) in macrophages. Furthermore, the RNA expression levels of *whiB3* and *sodA* (**B**,**D**) were determined in intracellular bacteria. The data show the maximums and minimums by quartiles. * *p* < 0.05 significant difference compared to the expression of uninfected macrophages (**A**,**C**) or intracellular bacteria without treatment (**B**,**D**), *n* = 4–8. The Wilcoxon test was used to compare to the control. In panels (**A**,**C**), the dashed horizontal line represents the median of the uninfected macrophages without treatment, and in panels (**B**,**C**), it represents the median of the intracellular bacteria without treatment.

**Figure 4 microorganisms-10-01821-f004:**
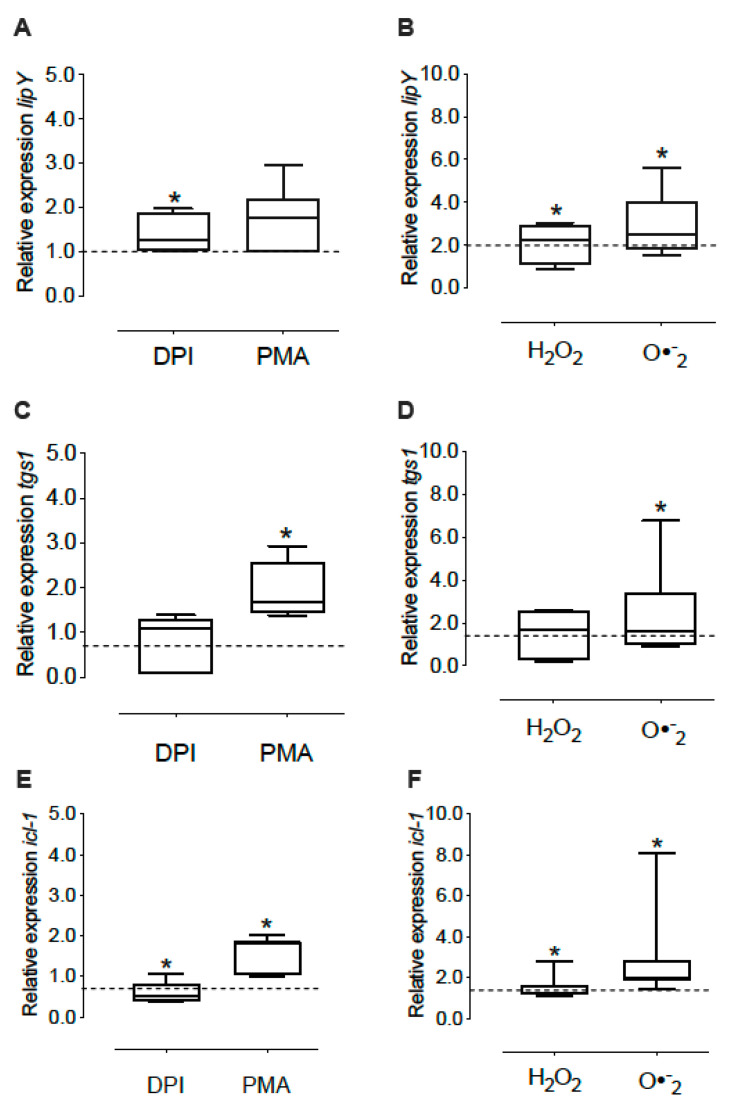
Expression of the *lip-Y*, *tgs-1* and *icl-1* in intracellular mycobacteria under oxidant conditions. The MDM were treated with NOX modulators (DPI and PMA) for 1 h or oxidizing agents (H_2_O_2_ and O_2_^•−^) for 15 min, and then infected with *M. tuberculosis* H37Ra (MOI 1:15) in presence of these oxidants for 1 h. Non-phagocyted bacteria were removed, and fresh medium containing NOX modulators or oxidants were added for 2 h. Then, we determined the expression of *lip-Y* (**A**,**B**), *tgs-1* (**C**,**D**) and *icl-1* (**E**,**F**). The data show the maximums and minimums by quartiles. * *p* < 0.05 significant difference compared to intracellular bacteria without treatment (control), *n* = 7–8. The dashed horizontal line represents the median of the control, and the Wilcoxon test was used to compare against the control (without treatment).

**Figure 5 microorganisms-10-01821-f005:**
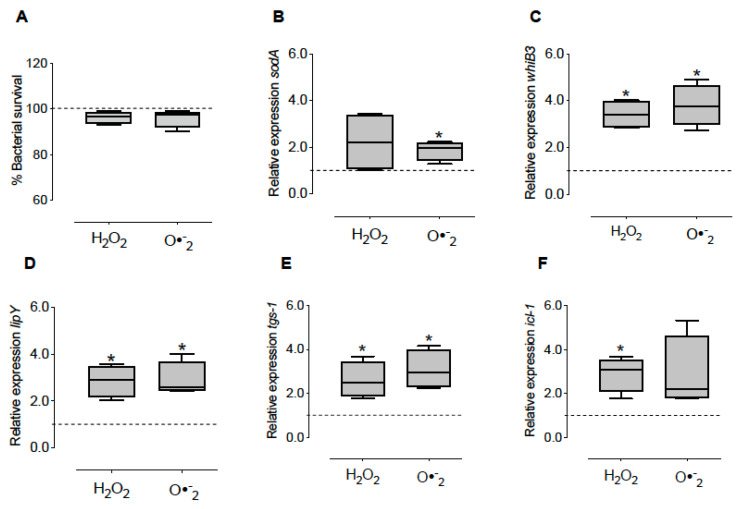
Effect of oxidizing agents on bacterial survival and gene expression of *M. tuberculosis*. Bacteria were grown in the presence of 5 mM H_2_O_2_ and O_2_^•−^ generator system for 2 h. Then, we determined the survival (**A**) and the expression of *sodA* (**B**), *whiB3* (**C**), lipY (**D**), *tgs-1* (**E**) and *icl-1* (**F**). The data show the maximums and minimums by quartiles. * *p* < 0.05 significant difference compared to the expression of the control, *n* = 4. The dashed horizontal line represents the median of the control, and the Wilcoxon test was used to compare against the control (untreated mycobacterial culture).

**Figure 6 microorganisms-10-01821-f006:**
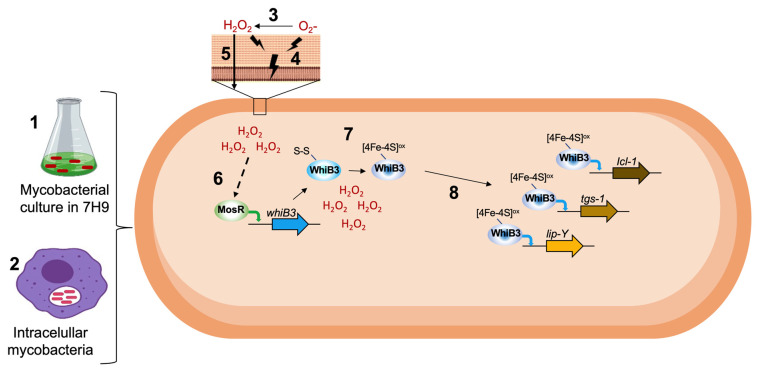
Effect of oxidants on gene expression of free-living mycobacteria cultured in 7H9 (1) and intracellular mycobacteria (2). The superoxide anion (O_2_^●^^−^) is converted into hydrogen peroxide (H_2_O_2_) (3). Both oxidants damage the cell wall and membrane, generating lipoperoxidation (4). H_2_O_2_ is an electrically neutral molecule that crosses membranes (5), increasing its concentration inside the cell causing an imbalance in the redox state. The MosR regulator senses oxidative stress and induces whiB3 expression (6). Oxidative conditions favor the [4Fe-4S] cluster oxidation of WhiB3 (7). [4Fe-4S]ox-WhiB3 can bind to promoters of its target genes (8), generating increased expression of *lip-Y*, *tgs-1*, and *icl-1*, causing a metabolic change that leads to a state of dormancy.

**Table 1 microorganisms-10-01821-t001:** Primers used for gene expression analysis.

Target	Sequences	Reference
*whiB3*	F: tggactcatcgatgttcttcc	This work
	R: tagggctcaccgacctctaa	
*lip-Y*	F: gtattagccgctgccgagga	[30]
	R: gataccgctggcgaattcactct	
*tgs-1*	F: aacgaagaccagttattcgagc	[30]
	R: ctcatactttcatcggagagcc	
*icl-1*	F: cggatcaacaacgcactgca	[30]
	R: ttctgcagctcgtagacgtt	
*sodA*	F: acaccttgccagacctgga	This work
	R: cgccctttacgtaggtggc	
*rRNA* 16S	F: ggtgcgagcgttgtccgg	This work
	R: cgcccgcacgctcacagtta	

**Table 2 microorganisms-10-01821-t002:** Putative MosR binding site in the *whiB3* promoter region of *M. tuberculosis*.

Location	Promotor	Sequence *
−25	*MosR* (*Rv1049*)	atacg tgtag ctaca cgagc
−91	*WhiB3* (*Rv3416*)	ttagg cgtac tcaca gcatg
	consensus sequence	g tgtan ntaca c

* Alignment of the MosR binding sequence (U-MosR) and consensus sequence described by Brugarolas [3] with the promoter region of *whiB3*.

## Data Availability

The data that support the findings of this study are available upon reasonable request from the corresponding author.

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
