# Peer review of "Mycobacterium tuberculosis whiB3 and Lipid Metabolism Genes Are Regulated by Host Induced Oxidative Stress"

_microorganisms, 2022, doi:10.3390/microorganisms10091821_

Round 1

Reviewer 1 Report

The authors managed to bring very important new information in the area. However, I have a few small points to make.

Isolation of monocytes from PBMCs using magnetic beads is a good strategy. However, it would be interesting to confirm the differentiation of monocytes into macrophages using specific antibodies, including the loss of CD14 expression. The same procedure applies to alveolar macrophages, since the evaluation only by size and granularity is not considered a good method for evaluating macrophage differentiation.

The legend of figure 1 is not in the same pattern as the other figures. One should: mention the quartiles and inform the experimental N in the same way.

Line 259: M. tuberculosis should be in italics

In the analysis of ROS production and cell viability, I missed the controls where the cells are treated but not infected and also the cells without treatment or infection. It would be nice to include this data.

The results need to be better described, informing which letter each data is mentioning.

In figure 4, which macrophage was analyzed (MDM or AM)?

In figure 5, would the control be the bacteria without treatment? Include this information.

At the beginning of the caption of figure 5, there was a confusion in the writing: "Expression Effect..."

Only in Figure 1 was used alveolar macrophage. What is the reason? It would be interesting to make this same parallel with the two cell types in all results, considering that in the context of Mtb infection, AM would be the closest to reality.

The data obtained in the manuscript are interesting, however, it would be important to carry out experiments using M. tuberculosis mutant for the whiB3 gene so that the results are more convincing and, thus, show the direct influence of HWhiB3 on the intracellular persistence of the bacterium in macrophages due to synthesis of reserve lipids.

In lines 351 and 365, there was an error in the bibliographic citations.

Reviewer 2 Report

This is a very interesting study with very clean methodologies. 

The results are properly concluded.

Question: Authors use the term lipid metabolism too casually. based on the results it should be either lipid catabolism or anabolism. Are authors concluding to say that M.tb use their lipid stores as an energy source to survive hostile and limiting intracellular stages? Authors need to add salient features and translational aspects in their discussion and conclusion. Also though not tested these results can be compared with dormancy revival of these pathogens and changes in lipid contents as reported by different workers. 

This is a significant finding and the manuscript needs a better and broader discussion. 

Reviewer 3 Report

In this manuscript the authors demonstrate the upregulation of whiB3 under conditions of oxidative stress in vitro and in intracellular Mtb. They hypothesize that whiB3 upregulation leads to Mtb transition into dormancy and indicate that the presence of a mosR binding site in the whiB3 promoter may explain whiB3 regulation during oxidative stress. Following are the concerns:    

Major

1.     The mosR hypothesis is interesting and easily testable in the same experimental samples used throughout this paper. The authors should check the transcript levels of mosR, since it is autoregulated, in H2O2 and O2 treated samples as well as in intracellular Mtb samples to consolidate their hypothesis of mosR dependent regulation of whiB3. 

2.    Minor

Line 273 “oxidizing”

Line 362 “tgs-1”

Line 375 “the absence of whiB3 was…”

Round 2

Reviewer 1 Report

The authors made the necessary changes and/or justifications. Therefore, I consider the article suitable for publication.